# Lie prevalence, lie characteristics and strategies of self-reported good liars

**Brianna L. Verigin**[1,2]*, **Ewout H. Meijer**[1], **Glynis Bogaard**[1], **Aldert Vrij**[2]

**1** Forensic Psychology Section, Faculty of Psychology and Neuroscience, Maastricht University, Maastricht, Netherlands, **2** Department of Psychology, University of Portsmouth, Portsmouth, United Kingdom

* brianna.verigin@maastrichtuniversity.nl

**Data Availability Statement:** The data underlying the results presented in the study are available from https://doi.org/10.17605/OSF.IO/GB4RP.

**Funding:** This research was supported by a fellowship awarded from the Erasmus Mundus

## Abstract

Meta-analytic findings indicate that the success of unmasking a deceptive interaction relies more on the performance of the liar than on that of the lie detector. Despite this finding, the lie characteristics and strategies of deception that enable good liars to evade detection are largely unknown. We conducted a survey ($n$ = 194) to explore the association between lay-people's self-reported ability to deceive on the one hand, and their lie prevalence, characteristics, and deception strategies in daily life on the other. Higher self-reported ratings of deception ability were positively correlated with self-reports of telling more lies per day, telling inconsequential lies, lying to colleagues and friends, and communicating lies via face-to-face interactions. We also observed that self-reported good liars highly relied on verbal strategies of deception and they most commonly reported to i) embed their lies into truthful information, ii) keep the statement clear and simple, and iii) provide a plausible account. This study provides a starting point for future research exploring the meta-cognitions and patterns of skilled liars who may be most likely to evade detection.

## Introduction

Despite the importance of being able to detect deception, research has consistently found that people are unable to do so. In fact, the accuracy rates vary around chance level [1, 2]. Lacking good lie detectors, a growing body of evidence indicates that the accuracy of detecting deception depends more on the characteristics of the liar and less on the lie detection ability of the judge [3–7]. The meta-analysis of Bond and DePaulo [3] provided robust evidence that liars vary in their detectability. Their analysis showed that differences in detectability from sender to sender are more reliable than differences in credulity from judge to judge, with reliability coefficients of .58 and .30, respectively. This pattern of results was replicated by other researchers [5], lending support to the proposition that liar characteristics exert a powerful influence on lie detection outcomes. Moreover, it has been shown that sender demeanour explained up to 98% of the variance in detection accuracy [7].

Yet, only a handful of studies have attempted to determine individual differences in the ability to lie credibly [8–12]. Research on what characterizes those who escape detection, i.e., good liars, would be highly beneficial in investigative settings. Thus, a focus on the liar, in

Joint Doctorate Program The House of Legal Psychology (EMJD-LP) with Framework Partnership Agreement (FPA) 2013-0036 and Specific Grant Agreement (SGA) 2016-1339 to Brianna L. Verigin. The funders had no role in study design, data collection and analysis, decision to publish, or preparation of the manuscript.

**Competing interests:** The authors have declared that no competing interests exist.

particular the skilled liar, was the aim of this study. Specifically, the present manuscript reports an exploratory study addressing how self-reported deception ability is associated with lie prevalence and lie characteristics, and how self-reported good liars utilize strategies for deceiving.

First, we investigated the relationship between liars' self-reported lie-telling frequency and self-reported deception ability. The most widely cited research on deception prevalence estimates the frequency at an average of once or twice per day [13, 14]. More recent research, however, shows that the distribution of lies per day is considerably skewed. The majority of lies are told by only a handful of prolific liars [15–17]. Specifically, in a survey of nearly 3,000 participants, researchers found that 5% of respondents accounted for over 50% of all the lies reported within the past 24 hours, whereas the majority of subjects reported telling no lies at all [15]. Several additional studies, as well as a reanalysis of DePaulo et al.'s [13] diary study, have validated that the majority of lies are told by a minority of people [14, 16]. These few prolific liars tend to tell more serious lies that carry significant consequences if detected [15]. Also, people who self-reported to lie more often were more prone to cheating in laboratory tasks for personal profit [17]. It is possible that these prolific liars also perceive themselves as more skilled at deceiving and tell more lies that they think will stay undetected, either because they believe the receiver will not try to find out or they believe they are good enough to fool the receiver.

Second, our investigation examined whether characteristics of lies differ as a function of self-reported deception ability. The first of these characteristics is the type of lie. This can refer to the severity; at one end are white lies, which are relatively common [18] and often used to ease social interactions (e.g., telling your mother-in-law that her baking is delicious when you actually dislike sweets) [19], while at the other end are bold-faced fabrications, which are less common and typically serve to protect the liar (e.g., denying having had an affair) [20]. The latter type of lies are also encountered more by the legal system [21]. Other taxonomies of lies also exist, for example lies of omission or lies embedded into the truth; however, research has yet to explore how the types of lies could differ as a function of deception ability. Is it that, for example, good liars tend to utilize a certain type of lie which facilitates their success? The second characteristic is the receiver of the lie. Lies can be communicated to a variety of individuals ranging from family, romantic partners, and friends to strangers, colleagues, or authority figures. Previous research has shown that people lie less frequently in close relationships than in casual relationships [22]. A third characteristic we are interested in is the medium of deception, as this can also influence the success of one's lie. Some liars, for instance, prefer online communication [23]. This would fit the liars' (erroneous) belief their deception will leak out via behavioural cues [24]. It is unknown, however, if or how good liars concentrate their lies to specific individuals or communicate via certain mediums.

Finally, we examined how self-reported good liars utilize strategies for deceiving. The idea that liars adopt strategies to enhance the likelihood of successfully deceiving stems from research on impression and information management. Both forms of regulation relate to the idea that much of social behaviour is controlled for the purpose of interpersonal presentation [25, 26]. In legal contexts, both liars and truth tellers are motivated to achieve a favourable impression and attempt to do so by regulating their speech and behaviour, albeit liars more so than truth tellers [27]. The topic of deceivers' strategies has received some empirical attention [27–30]. For example, it was found that among the principal strategies of criminal offenders were "Staying close to the truth," and "Not giving away information" [31]. Researchers have also capitalized on this increased awareness of liars' and truth tellers' strategies by developing strategy-based lie detection tools. For instance, the Verifiability Approach (VA) [32, 33] exploits liars' strategy of providing detailed statements that are embellished with unverifiable information. Moreover, some researchers have speculated that good liars might use effective strategies to conduct their behaviour, by attempting to act in line with people's beliefs about

how truth tellers behave while avoiding behaviour associated with liars [10]. Still, surveying expert liars about their strategies as a source of insight into real-world deception remains a highly underdeveloped research avenue [34].

## Materials and methods

This study was approved by the ethical committee of the Faculty of Psychology and Neuroscience at Maastricht University. Participants read and signed the informed consent in accordance with the Declaration of Helsinki.

### Participants

The sample consisted of 194 participants (97 females; 95 males; 2 preferred not to say; $M_{age}$ = 39.12 years, $SD_{age}$ = 11.43) recruited via Amazon Mechanical Turk (mTurk). Most participants reported being U.S. citizens (*n* = 175), whereas the remainder (*n* = 19) reported Indian citizenship. Participants who completed the study were paid 1.75 USD. Participants could participate in the study if they reported to be able to understand and write English at an advanced level. To ensure data quality, participants were required to have the mTurk Masters Qualification that is awarded to those who have demonstrated continual excellence across a wide range of mTurk projects. An additional 133 participants began the questionnaire but did not complete it, therefore their data were discarded. Data from nine participants were also removed because of insufficient responses. We reached our sample size (*n* = 194) after these exclusions. The study was approved by the standing ethical committee.

### Procedure

The online questionnaire was created on Qualtrics online platform. After providing informed consent, participants were provided definitions of lying and deception modelled from previous research [2, 13; see Supporting Information]. Participants were asked to read these definitions carefully and to consider them while making responses throughout the questionnaire. In the first part of the questionnaire, participants reported their experience with telling lies in daily life. Participants rated on a 10-point Likert scale (1 –very poor to 10 –excellent) "How good are you at successfully deceiving others (i.e., getting away with lies)?" Next, they reported the estimated number of lies told during the past 24 hours. Participants then responded to multiple-response questions about i) the types of lies told during the past 24 hours (options: white lies, exaggerations, lies of omission/concealment, lies of commission/fabrications, embedded lies; see the Supporting Information for the definitions provided to participants); ii) the receivers of their deception (options: family, friend, employer, colleague, authority figure, or other); and iii) the mediums of their deception (options: face-to-face, over the phone, social media, text message, email, or other).

The second part of the questionnaire probed the deceiver's strategies. Participants provided an open-ended response to explain "In general, what strategy or strategies do you use when telling lies?" They were then asked to rate on a 10-point Likert scale (1 –not important to 10 –very important) how important they consider verbal strategies of deception and nonverbal strategies of deception to be for getting away with lies (for the definitions provided to participants, see the Supporting Information). Finally, participants indicated which verbal strategies they use when telling lies in general from a pre-determined set (options: reporting from previous experience, providing details the person cannot check [i.e., unverifiable details], telling a plausible story, etcetera). The options included in this list were drawn from empirical findings regarding liars' strategies and cues to deception [26, 32, 35]. Participants then provided demographic information regarding their age, sex, citizenship, ethnicity and education. We explored

the association between laypeople's self-reported deception ability and their sex and education level. Finally, an additional part of the questionnaire asked participants to recall a time in which they had told a serious lie and to report their rationale for lying and their strategies. We examined how the deception rationale influenced their motivation, preparation, strategies, and perceived success of the lie. To conserve manuscript length, the final section of the questionnaire is reported in the Supporting Information.

**Qualitative analysis.** To code the participants' self-reported strategies into data-driven categories, the first author performed a content analysis on the open-ended responses to the question probing their use of strategies. First, each participant's strategy or strategies was identified, then all overlapping responses were combined, and these strategies were condensed into several dominant categories with theoretical similarities (i.e., relating to behavioural control or verbal control, etcetera). The main coder completed each stage of this process and all authors approved upon the final categories. Seven categories emerged from this coding method, for example omitting certain information, relating to truthful information, or controlling behaviour (see Table 1).

To establish inter-rater reliability, the main coder and a second coder coded a randomly selected 20% of the participants' open-ended responses into the appropriate categories. A two-way mixed effects model measuring consistency [36] showed that raters were highly consistent across all categories (Single Measures *ICCs* ranged from .79 to 1.00). After confirming that the raters were consistent, the main coder (first author) completed the remaining sample of participant responses and only these scores were used in the analysis.

## Results

We were interested in i) replicating previous findings regarding the distribution skewness of lie-telling frequency and exploring how these patterns relate to self-reported deception ability; ii) isolating lie characteristics as a function of deception ability; and iii) exploring the strategies of deception used by self-reported good liars.

### Lie prevalence and characteristics

We investigated how laypeople lie in daily life by examining the frequency of lies, types of lies, receivers and mediums of deception within the past 24 hours. Overall, participants indicated telling a mean of 1.61 lies during the last 24 hours (*SD* = 2.75; range: 0–20 lies), but the distribution was non-normally distributed, with a skewness of 3.90 (*SE* = 0.18) and a kurtosis of 18.44 (*SE* = 0.35). The six most prolific liars, less than 1% of our participants, accounted for 38.5% of the lies told. Thirty-nine percent of our participants reported telling no lies. Fig 1 displays participants' lie-telling prevalence.

Participants' endorsement of the type, recipient, and medium of their lies are shown in Fig 2. Participants mostly reported telling white lies, to family members, and via face-to-face interactions. All lie characteristics displayed non-normal distributions (see the Supporting Information for the complete description).

**Lie prevalence and characteristics as a function of deception ability.** Next, we conducted correlational analyses to examine the association of our participants' lie frequency and characteristics with their self-reported deception ability. An increase in self-reported ability to deceive was positively correlated to a greater frequency of lies told per day, $r(192)$ = .22, $p$ = .002, and with higher endorsement of telling white lies and exaggerations within the last 24 hours ($r(192)$ = .16, $p$ = .023 and $r(192)$ = .16, $p$ = .027, respectively). There were no significant associations between self-reported deception ability and reported use of embedded lies, $r(192)$ = .14, $p$ = .051; lies of omission, $r(192)$ = .10, $p$ = .171; or lies of

**Table 1. Endorsement of general qualitative deception strategies and descriptive statistics as a function of deception ability.**

| Interview Strategies | N | M | SD | $X^2$ |
|---|---|---|---|---|
| Omitting certain information | 76 | 0.39 | 0.49 | $\chi^2(2) = 3.00$, $p = .223$, $V = .124$ |
| Poor | 25 | 0.49 | 0.51 | |
| Neutral | **28** | **0.37** | **0.49** | |
| Good | 23 | 0.34 | 0.48 | |
| Providing certain information | 49 | 0.25 | 0.44 | $\chi^2(2) = 5.49$, $p = .064$, $V = .168$ |
| Poor | 7 | 0.14 | 0.35 | |
| Neutral | 20 | 0.27 | 0.45 | |
| Good | **22** | **0.32** | **0.47** | |
| Relating to truthful information | 49 | 0.25 | 0.44 | $\chi^2(2) = 5.02$, $p = .081$, $V = .161$ |
| Poor | 7 | 0.14 | 0.35 | |
| Neutral | **23** | **0.31** | **0.46** | |
| Good | 19 | 0.28 | 0.45 | |
| Behavioural control | 39 | 0.20 | 0.40 | $\chi^2(2) = 2.69$, $p = .260$, $V = .118$ |
| Poor | 9 | 0.18 | 0.39 | |
| Neutral | 12 | 0.16 | 0.37 | |
| Good | **18** | **0.26** | **0.44** | |
| Miscellaneous strategies | 44 | 0.23 | 0.42 | $\chi^2(2) = 1.29$, $p = .524$, $V = .082$ |
| Poor | 9 | 0.18 | 0.39 | |
| Neutral | 17 | 0.23 | 0.42 | |
| Good | **18** | **0.26** | **0.44** | |
| No strategy | 10 | 0.05 | 0.22 | $\chi^2(2) = 8.26$, $p = .016$, $V = .206$ |
| Poor | **6** | **0.12** | **0.33** | |
| Neutral | 4 | 0.05 | 0.23 | |
| Good | 0 | 0 | 0 | |
| Not Applicable | 15 | 0.08 | 0.27 | $\chi^2(2) = 1.23$, $p = .540$, $V = .080$ |
| Poor | 4 | 0.08 | 0.27 | |
| Neutral | 4 | 0.05 | 0.23 | |
| Good | 7 | **0.10** | **0.31** | |

*Note.* The *N* column represents the number of participants who endorsed each strategy, both in the total sample and for Poor, Neutral and Good liars, respectively. The total number of endorsed strategies surpasses the sample size of 194 because each participant could report multiple strategies that may have fallen into more than one category. The bolded numbers represent the group with the highest endorsement of each strategy.

commission, $r(192) = .10$, $p = .161$. Higher self-reported deception ability was significantly associated with telling lies to colleagues, $r(192) = .27$, $p < .001$, friends, $r(192) = .16$, $p = .026$, and "other" receivers of deception, $r(192) = .16$, $p = .031$; however, there were no significant associations between self-reported ability to lie and telling lies to family, employers, or authority figures ($r(192) = .08$, $p = .243$; $r(192) = .04$, $p = .558$; and $r(192) = .11$, $p = .133$, respectively). Finally, higher values for self-reported deception ability were positively correlated to telling lies via face-to-face interactions, $r(192) = .26$, $p < .001$. All other mediums of communicating the deception were not associated with a higher reported ability, as follows: Via phone conversations, text messaging, social media, email, or "other" sources ($r(192) = .13$, $p = .075$; $r(192) = .13$, $p = .083$; $r(192) = .03$, $p = .664$; $r(192) = .05$, $p = .484$; $r(192) = .10$, $p = .153$, respectively).

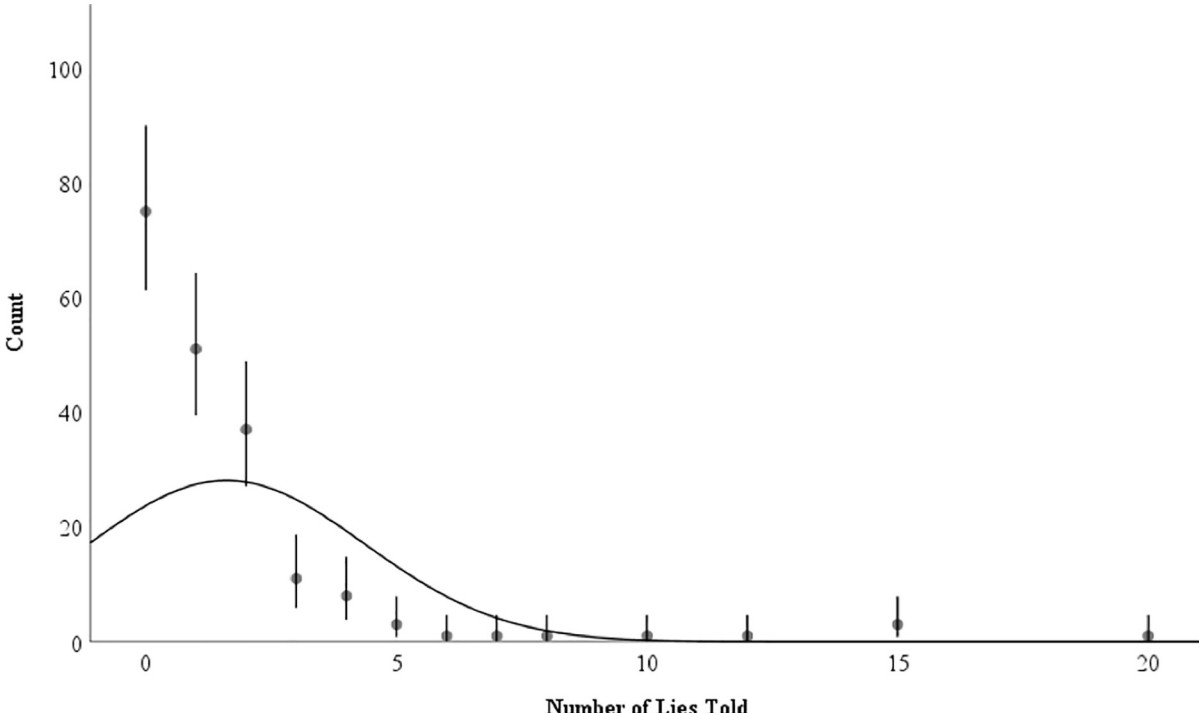

**Fig 1. Scatterplot of participants' self-reported lie-telling frequency during the past 24 hours.** The distribution curve represents the mean and standard deviation of the total sample. Error bars represent 95% confidence intervals.

### Deception strategies of good liars

We were also interested in exploring the strategies of deception, particularly those of good liars. To test this, we created categories representing participants' self-reported deception ability, using their scores from the question asking about their ability to deceive successfully, as follows: Scores of three and below were combined into the category of "Poor liars" ($n$ = 51); scores of 4, 5, 6, and 7 were combined into the category of "Neutral liars" ($n$ = 75); and scores of eight and above were combined into the category of "Good liars" ($n$ = 68).

Table 1 provides an overview of the exact values regarding the endorsement of each deception strategy that emerged from the qualitative coding. To examine whether there were associations between the reported strategies and varying deception abilities, we conducted a series of chi square tests of independence on participants' coded responses to the question regarding their general strategies for deceiving. We did not observe any statistically significant associations between self-reported deception ability and the endorsement of any strategy categories (see Table 1), apart from one exception. We observed a significant association between Poor, Neutral and Good liars and the endorsement of using "*No strategy*". Pairwise comparisons were performed using Dunn's procedure [37] with a corrected alpha level of .025 for multiple tests. This analysis revealed a significant difference in endorsing "*No strategy*" only between the Good and Poor liars, $p$ = .004. However, we did not meet the assumption of all expected cell frequencies being equal to or greater than five and as such these data may be skewed. Based on Cohen's guidelines [38], all associations were small to moderate (all Cramer's $Vs$ < .206).

**Verbal and nonverbal strategies.**   To investigate whether participants differed in their endorsement of the importance of verbal versus nonverbal strategies based on their self-

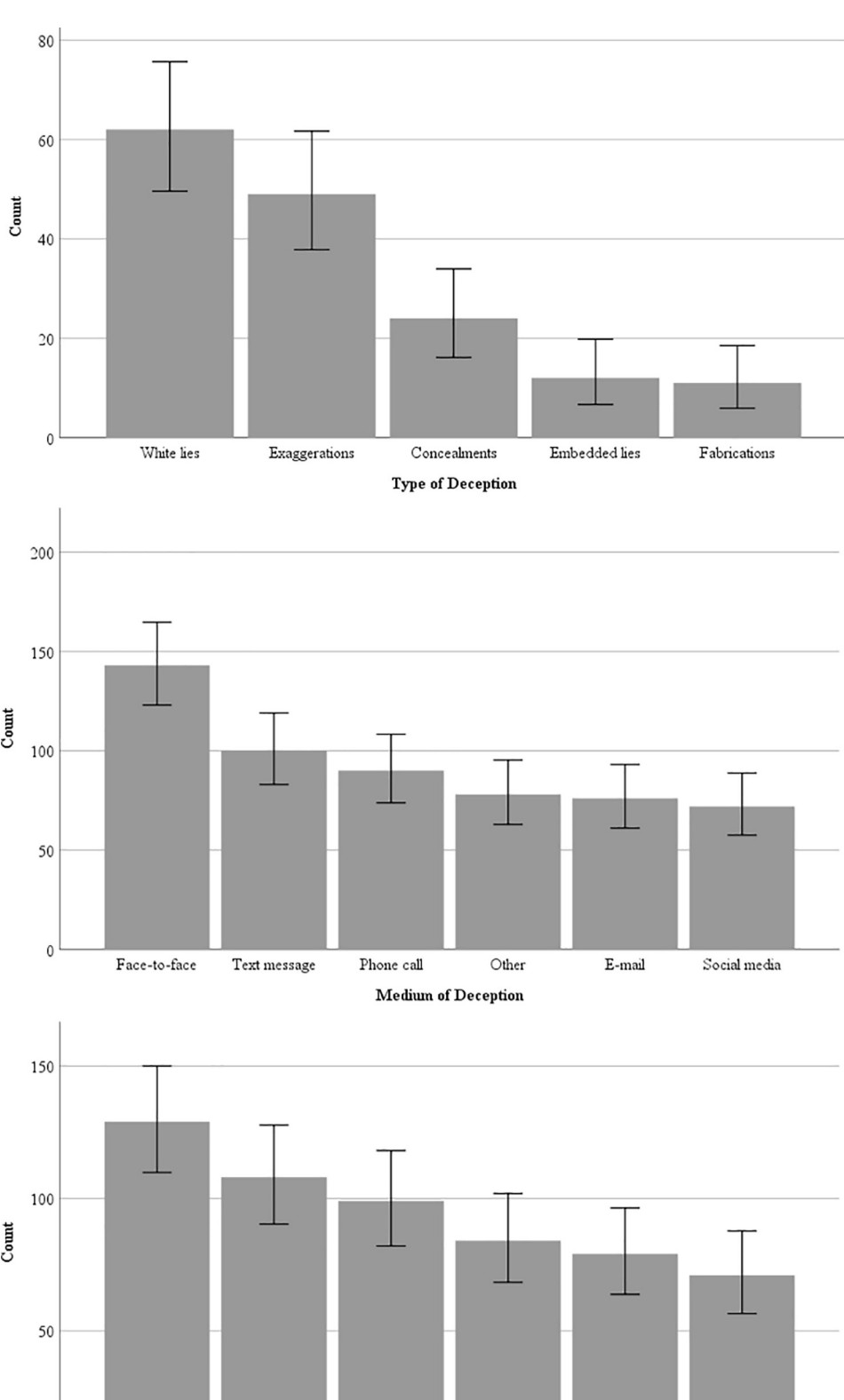

**Fig 2. Bar charts displaying the frequency of the types, receivers, and mediums of deception endorsed by participants for their reported lies during the past 24 hours.** Error bars represent 95% confidence intervals. For deception recipients, "other" refers to individuals such as intimate partners or strangers; for deception mediums, "other" refers to online platforms not included in the provided list.

reported deception ability, we conducted two between-subjects ANOVAs with deception ability (Poor, Neutral, Good) on participants' Likert scale ratings of the importance of verbal and nonverbal strategies. Additionally, the data were examined by calculating Bayesian ANOVAs with default prior scales, using JASP software. We report the Bayesian factors [*BF*; see 39, 40] in line with the guidelines by Jarosz and Wiley [39], adjusted from Jeffreys [41]. For ease of interpretation, $BF_{10}$ is used to indicate the Bayes factor as evidence in favour of the alternative hypothesis, whereas $BF_{01}$ is used to indicate the Bayes factor as evidence in favour of the null hypothesis.

First, we found a significant effect of self-reported deception ability on participants' endorsement of verbal strategies, $F(2, 191) = 5.62$, $p = .004$, $\eta_P^2 = .056$; $BF_{10} = 7.11$. Post hoc comparisons indicated that Good liars rated verbal strategies as significantly more important than Neutral liars ($M_{\text{diff}} = -0.82$, 95% CI [-1.47, -0.18], $p = .009$), and Poor liars ($M_{\text{diff}} = -0.83$, 95% CI [-1.54, -0.11], $p = .018$). Participants across groups did not differ with respect to their endorsement of the importance of nonverbal strategies, $F(2, 191) = .003$, $p = .997$, $\eta_P^2 < .001$; $BF_{01} = 18.55$.

Next, we examined which specific verbal strategies participants reported to use when lying. We asked participants to indicate, from a list of ten options, which strategies they use. Table 2 provides an overview of the strategies endorsed by Poor, Neutral, and Good liars. Across all groups, the most frequently reported strategies were "*Keeping the statement clear and simple*" (endorsed by 17.6% of participants), "*Telling a plausible story*" (15.1% of participants), "*Using avoidance/being vague about details*" (13.2% of participants) and "*Embedding the lie into an otherwise truthful story*" (13.1% of participants). To examine differences in the endorsement of the strategies across Poor, Neutral, and Good liars we conducted a series of one-way between-subjects ANOVAs. Significant differences emerged for eight of the strategies, as follows: "*Embedding the lie*," $F(2, 191) = 11.97$, $p < .001$, $\eta_P^2 = .111$; $BF_{10} = 1438.20$; "*Matching the amount of details in the deceptive component of the statement to the truthful component*," $F(2, 191) = 4.77$, $p = .010$, $\eta_P^2 = .048$; $BF_{10} = 3.32$; "*Matching the type of details of the deceptive component of the statement to the truthful component*," $F(2, 191) = 3.56$, $p = .030$, $\eta_P^2 = .036$; $BF_{10} = 1.15$; "*Keeping the statement clear and simple*," $F(2, 191) = 5.07$, $p = .007$, $\eta_P^2 = .050$; $BF_{10} = 4.15$; "*Telling a plausible story*," $F(2, 191) = 5.48$, $p = .005$, $\eta_P^2 = .054$; $BF_{10} = 5.98$; "*Providing unverifiable details*," $F(2, 191) = 4.95$, $p = .008$, $\eta_P^2 = .049$; $BF_{10} = 3.78$, and "*Avoidance*," $F(2, 191) = 3.79$, $p = .024$, $\eta_P^2 = .038$; $BF_{10} = 1.43$. Interestingly, Good liars reported using all of the above strategies significantly more than Poor liars (all *p*'s < .025). The only exception was that Poor liars reported using the avoidance strategy significantly more than Good liars ($p = .026$). Finally, there were no significant differences between Good, Neutral, and Poor liars in endorsing "*Reporting from previous experience/memory*" ($F(2, 191) = 1.32$, $p = .268$, $\eta_P^2 = .014$; $BF_{01} = 5.96$), "*Using complete fabrication*" ($F(2, 191) = 0.57$, $p = .565$, $\eta_P^2 = .006$; $BF_{01} = 11.36$), and "*Using other strategies*" ($F(2, 191) = 0.51$, $p = .600$, $\eta_P^2 = .005$; $BF_{01} = 11.96$). See Table 2 for the exact values and applicable post hoc comparisons.

## Exploratory testing of liar characteristics

Finally, we also explored the associations between sex and education level and laypeople's self-reported deception ability by conducting a series of chi square tests of association. We

**Table 2. Endorsement of predetermined deception strategies and descriptive statistics as a function of deception ability.**

| Interview Strategies | N | M | SD | Bonferroni Comparisons | |
|---|---|---|---|---|---|
| | | | | **Poor** | **Neutral** |
| Keeping the statement clear and simple | 112 | | | | |
| Poor | 20 | 0.39 | 0.49 | | |
| Neutral | 49 | 0.65 | 0.48 | **.010** | |
| Good | 43 | 0.63 | 0.49 | **.025** | 1.00 |
| Telling a plausible story | 96 | | | | |
| Poor | 17 | 0.33 | 0.48 | | |
| Neutral | 36 | 0.48 | 0.50 | .302 | |
| Good | 43 | 0.63 | 0.49 | **.004** | .195 |
| Avoidance | 84 | | | | |
| Poor | 28 | 0.55 | 0.50 | | |
| Neutral | 35 | 0.47 | 0.50 | 1.00 | |
| Good | 21 | 0.31 | 0.47 | **.026** | .167 |
| Embedding the lie | 83 | | | | |
| Poor | 13 | 0.26 | 0.44 | | |
| Neutral | 26 | 0.35 | 0.48 | .850 | |
| Good | 44 | 0.65 | 0.48 | **< .001** | **< .001** |
| Providing unverifiable details | 76 | | | | |
| Poor | 12 | 0.24 | 0.43 | | |
| Neutral | 29 | 0.39 | 0.49 | .251 | |
| Good | 35 | 0.52 | 0.50 | **.006** | .338 |
| Matching the *type* of details between lies and truths | 71 | | | | |
| Poor | 12 | 0.24 | 0.43 | | |
| Neutral | 27 | 0.36 | 0.48 | .453 | |
| Good | 32 | 0.47 | 0.50 | **.025** | .503 |
| Reporting from previous experience | 55 | | | | |
| Poor | 10 | 0.20 | 0.40 | | |
| Neutral | 23 | 0.31 | 0.46 | .535 | |
| Good | 22 | 0.32 | 0.47 | .387 | 1.00 |
| Matching the *amount* of details between lies and truths | 38 | | | | |
| Poor | 5 | 0.10 | 0.30 | | |
| Neutral | 12 | 0.16 | 0.37 | 1.00 | |
| Good | 21 | 0.31 | 0.47 | **.012** | .072 |
| Using complete fabrication | 14 | | | | |
| Poor | 2 | 0.04 | 0.20 | | |
| Neutral | 6 | 0.08 | 0.27 | 1.00 | |
| Good | 6 | 0.09 | 0.29 | .930 | 1.00 |
| Using other strategies | 7 | | | | |
| Poor | 3 | 0.06 | 0.24 | | |
| Neutral | 2 | 0.03 | 0.16 | 1.00 | |
| Good | 2 | 0.03 | 0.17 | 1.00 | 1.00 |

*Note.* The N represents the number of participants who endorsed each strategy per group. Post hoc comparisons were conducted with the Bonferroni correction, and the p-values are displayed in the table. The bolded numbers represent the significant cell comparisons.

observed a significant association between sex and deception ability, $\chi^2(2) = 12.31$, $p = .002$, $V = .253$. Further examination revealed that, of those who self-reported to be Poor liars, 70%

($n$ = 35) were female compared to 30% ($n$ = 15) male. Additionally, of those who identified themselves as Good liars, 62.7% ($n$ = 42) were male whereas 37.3% ($n$ = 25) were female. Both column proportions were significantly different at the .05 level. We did not observe a significant association between participants' education level and their self-reported deception ability, $\chi^2$(4) = 9.09, $p$ = .059, $V$ = .153. The complete analyses are presented in the Supporting Information to conserve manuscript length.

## Discussion

We found that self-reported good liars i) may be responsible for a disproportionate amount of lies in daily life, ii) tend to tell inconsequential lies, mostly to colleagues and friends, and generally via face-to-face interactions, and iii) highly rely on verbal strategies of deception, most commonly reporting to embed their lies into truthful information, and to keep the statement clear, simple and plausible.

### Lie prevalence and characteristics

First, we replicated the finding that people lie, on average, once or twice per day, including its skewed distribution. Nearly 40% of all lies were reported by a few prolific liars. Furthermore, higher self-reported ratings of individuals' deception ability were positively correlated with self-reports of: i) telling a greater number of lies per day, ii) telling a higher frequency of white lies and exaggerations, iii) telling the majority of lies to colleagues and friends or others such as romantic partners, and iv) telling the majority of lies via face-to-face interactions. Importantly, skewed distributions were also observed for the other lie characteristics, suggesting that it may be misleading to draw conclusions from sample means, given that this does not reflect the lying behaviours of the average person. A noteworthy finding is that prolific liars also considered themselves to be good liars.

The finding that individuals who consider themselves good liars report mostly telling inconsequential lies is somewhat surprising. This deviates from the results of a previous study, which showed that prolific liars reported telling significantly more serious lies, as well as more inconsequential lies, compared to everyday liars [15]. However, small, white lies are generally more common [18] and people who believe they can get away with such minor falsehoods may be more inclined to include them frequently in daily interactions. It is also possible that self-reported good liars in our sample had inflated perceptions of their own deception ability because they tell only trivial lies versus lies of serious consequence.

Regarding the other lie characteristics, we found a positive correlation between self-reported deception ability and telling lies to colleagues, friends and others (e.g., romantic partners). This variation suggests that good liars are perhaps less restricted in who they lie to, relative to other liars who tell more lies to casual acquaintances and strangers than to family and friends [22]. Our results also showed that good liars tended to prefer telling lies face-to-face. This fits the findings of one of the only other studies to examine characteristics of self-reported good versus poor liars, which found that self-perceived good liars most commonly lied via face-to-face interactions versus through text chat [42]. This could be a strategic decision to deceive someone to their face, since people may expect more deception via online environments [43]. As researchers continue to examine the nature of lying and to search for ways of effectively detecting deception, it is important to recognize how certain lie characteristics may influence individuals' detectability as liars.

### Deception strategies

We also isolated the lie-telling strategies of self-reported good liars. People who identified as good liars placed a higher value on verbal strategies for successfully deceiving. Additional

inspection of the verbal strategies reported by good liars showed that commonly reported strategies were embedding lies into truthful information and keeping their statements clear, simple and plausible. In fact, good liars were more likely than poor liars to endorse using these strategies, as well as matching the amount and type of details in their lies to the truthful part/s of their story, and providing unverifiable details. A common theme among these strategies is the relation to truthful information. This fits with the findings of previous literature, that liars typically aim to provide as much experienced information as possible, to the extent that they do not incriminate themselves [35, 44]. Additionally, good liars used plausibility as a strategy for succeeding with their lies. This reflects the findings of the meta-analysis by Hartwig and Bond [45] that implausibility is one of the most robust correlates of deception judgements, and the results of DePaulo et al. [26] that one of the strongest cues to deception is liars' tendency to sound less plausible than truth tellers ($d$ = -0.23).

We also found that self-reported poor liars were more likely than good liars to rely on the avoidance strategy (i.e., being intentionally vague or avoiding mentioning certain details). Previous research suggests that this is one of the most common strategies used by guilty suspects during investigative interviews [46]. Additionally, all liars in our study expressed behavioural strategies as being important for deceiving successfully. This could be explained by the widespread misconceptions about the associations between lying and behaviour, for example that gaze aversion, increased movement or sweating are behaviours symptomatic of deception [2, 47].

There was inconsistency in our data between the responses to the qualitative strategy question and the multiple-response strategy question. Based on the qualitative strategy data it seems that Good, Neutral, and Poor liars do not differ in their use of strategies. However, robust differences emerged when we evaluated participants' endorsement of the predetermined strategies. One explanation for this finding is the difficulty people perceive when they have to verbalize the reasons for their behavior. Ericsson and Simon [48] suggest that inconsistencies can occur especially when the question posed is too vague to elicit the appropriate information, which might have been the case in our study. Another explanation for the discrepancy in the data between the two measurement procedures is that data-driven coding is inherently susceptible to human subjectivity, error, and bias [49, 50]. Such limitations apply to a lesser extent to coding based on predetermined categories that are derived from psychological theory, an approach which has been heavily used within the deception literature [2]. In any case, future research should continue exploring the deception strategies of good liars using a variety of methodological approaches. In particular, it would be beneficial to establish more reliable techniques for measuring interviewees' processing regarding their deception strategies. One potential idea could be to explore the effectiveness of using a series of cued questions to encourage the recall of specific aspects of interviewees' memory or cognitive processing. Another suggestion is to combine the data-driven and theory-driven approaches, whereby the coding system is generated inductively from the data but the coders draw from the theoretical literature when identifying categories [50].

## Limitations

Some methodological considerations should be addressed. First, the results of the present study are drawn from participants' self-reports about their patterns of deception in daily life. Sources of error associated with such self-report data limit our ability to draw strong inferences from this study. However, previous research has validated the use of self-report to measure lying prevalence by correlating self-reported lying with other measures of dishonesty [17]. Moreover, self-report data may not be as untrustworthy as critics argue, and in some

situations, it may be the most appropriate methodology [51]. This study was intended as an initial examination of the strategies and preferences of good liars, and surveying liars for their own perspectives provided a novel source of insight into their behaviour. A constraint to the generalizability of this research is that we did not establish the ground truth as to whether self-reported good liars are indeed skilled deceivers. Future research could attempt to extend our findings by examining deceivers' lie frequency, characteristics, and strategies after systematically testing their lie-telling ability within a controlled laboratory setting.

Second, one of the most frequent concerns about using Amazon MTurk relates to low compensation and resulting motivation [52, 53]. We took measures to ensure that our remuneration to participants was above the fair price for comparable experiments. Importantly, data collected through MTurk produces equivalent results as data collected from online and student samples [52, 54–58]. As well, mTurk surveys have been shown to produce a representative sample of the United States population that yields results akin to those observed from more expensive survey techniques, such as telephone surveys [57]. It speaks to the validity of our data, for example, that the self-reported prevalence of lies, and the endorsement of nonverbal deception strategies, replicates previous research. Nonetheless, the results of this study could be advanced if future research i) directly replicates our survey amongst different populations, for instance university students, and ii) conceptually replicates this research by evaluating different methodological approaches for measuring deception ability (e.g., via controlled evaluation) and good liars' strategies for deceiving (e.g., via cued recall).

## Implications and future research

This study explored the deception characteristics and strategies used by self-reported good liars. Deception researchers generally agree that the most diagnostic information is found in the content of liars' speech [59]. Content-based cues to deception, however, may be less effective for detecting good liars who rely highly on verbal strategies of deception. This could even offer an explanation for the modest effect sizes observed in the deception literature [60]. For instance, good liars in our study reported to strategically embed their lies into truthful information. This finding has potential implications for the reliability of credibility assessment tools that derive from the assumption that truth tellers' statements are drawn from memory traces whereas liars' statements are fabricated from imagination [61, 62]. If good liars draw on their memory of truthful previous experiences, then their statements may closely resemble those of their truth telling counterparts. Another interesting observation was that self-reported good liars were more likely than poor liars to provide unverifiable details. This fits with the findings of previous literature on the VA, which contends that liars provide information that cannot be verified to balance their goals of being perceived as cooperative and of minimizing the chances of falsification by investigators [32, 33]. A fruitful avenue of future research could be to further explore liars' strategic inclusion of truthful information and unverifiable details. Doing so may give lie detectors an advantage for unmasking skilled liars. It would also be interesting for future research to examine how good versus poor liars are affected by certain interview techniques designed to increase the difficulty of lying such as the reverse-order technique [63].

## Conclusion

In sum, this study yields new insights into the deception prevalence, characteristics, and strategies used by self-reported good liars. We replicated the finding that a minority of individuals account for the majority of lies told in daily life, and we provide evidence that these prolific liars also consider themselves good liars. We unveiled several lie characteristics of good liars:

They lean towards telling inconsequential lies, mostly to colleagues and friends, and generally via face-to-face interactions. Additionally, our results showed that self-reported good liars may attempt to strategically manipulate their verbal behaviour to stay close to the truth and to tell a plausible, simple, and clear story. This study provides a starting point for further research on the meta-cognitions and patterns of skilled liars, who may be more likely to evade detection in investigative settings.

## Supporting information

**S1 File. Questionnaire definitions: Deception and strategies.**
(DOCX)

**S2 File. Lie characteristics and distribution skewness.**
(DOCX)

**S3 File. Exploratory testing of liar characteristics.**
(DOCX)

**S4 File. Questionnaire part III: Recalling a serious lie.**
(DOCX)

## Author Contributions

**Conceptualization:** Brianna L. Verigin, Ewout H. Meijer, Glynis Bogaard.

**Data curation:** Brianna L. Verigin.

**Formal analysis:** Brianna L. Verigin.

**Funding acquisition:** Brianna L. Verigin.

**Investigation:** Brianna L. Verigin.

**Methodology:** Brianna L. Verigin.

**Project administration:** Brianna L. Verigin.

**Resources:** Brianna L. Verigin.

**Supervision:** Ewout H. Meijer.

**Visualization:** Brianna L. Verigin.

**Writing – original draft:** Brianna L. Verigin.

**Writing – review & editing:** Brianna L. Verigin, Ewout H. Meijer, Glynis Bogaard, Aldert Vrij.

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
