## [Decision Letter · Decision Letter 0]

17 Sep 2019

PONE-D-19-22731

Lie prevalence, lie characteristics and strategies of self-reported good liars

PLOS ONE

Dear Ms Verigin,

Thank you for submitting your manuscript to PLOS ONE. After careful consideration, we feel that it has merit but does not fully meet PLOS ONE’s publication criteria as it currently stands. Therefore, we invite you to submit a revised version of the manuscript that addresses the points raised during the review process.

We would appreciate receiving your revised manuscript by Nov 01 2019 11:59PM. To enhance the reproducibility of your results, we recommend that if applicable you deposit your laboratory protocols in protocols.io, where a protocol can be assigned its own identifier (DOI) such that it can be cited independently in the future. For instructions see: http://journals.plos.org/plosone/s/submission-guidelines#loc-laboratory-protocols

We look forward to receiving your revised manuscript.

Kind regards,

Giuseppe Sartori

Academic Editor

PLOS ONE

Journal Requirements:

2. Please ensure that you include a title page within your main document. You should list all authors and all affiliations as per our author instructions and clearly indicate the corresponding author.

Additional Editor Comments (if provided):

Reviewers' comments:

Reviewer's Responses to Questions

**Comments to the Author**

1. Is the manuscript technically sound, and do the data support the conclusions?

Reviewer #1: Yes

Reviewer #2: Yes

2. Has the statistical analysis been performed appropriately and rigorously? 

Reviewer #1: Yes

Reviewer #2: Yes

3. Have the authors made all data underlying the findings in their manuscript fully available?

Reviewer #1: Yes

Reviewer #2: No

4. Is the manuscript presented in an intelligible fashion and written in standard English?

Reviewer #1: Yes

Reviewer #2: Yes

5. Review Comments to the Author

Reviewer #1: The submitted manuscript describes a data-driven piece of scientific research about the prevalence of lies in everyday life, providing an exploratory analysis about lies characteristics and strategies of self-reported good liars.

The manuscript is written in a good quality standard English. The study is clear and well presented. The authors exposed and motivated the research questions addressed, each research question has been faced separately into the results section and, finally, the main findings were described considering what is known in the field. Furthermore, the main limitations of the study and their possible mitigations were exposed and the contradictions with previous findings on the subject were discussed and explained.

Concerning the data collection procedure, the authors seem aware of the limitations of using crowd-sourcing platforms in scientific research and the strategies and constraints employed seem enough for ensuring the data quality. Nonetheless, as stated by the authors, the inconsistencies emerged between the response to the qualitative strategy question and the multiple-response strategy question could be due to inattentive or careless responding by participants (a common issue when collecting data through crowd-sourcing platforms such as mTurk). In order to exclude this effect, a possible solution could be to collect an additional small sample in a more controlled setting (i.e. in the laboratory) and comparing the new results with the ones reported in this version of the manuscript.

Among the described findings, the self-reported good liars seem mostly telling inconsequential lies (white lies) and placing a higher value on verbal strategies than on behavioural ones for successfully deceiving. However, the results described in the supplementary materials concerning the last section of the questionnaire (where participants were asked which kind of strategies they use when telling serious lies) highlighted that "Behavioural manipulation" was the most frequently endorsed strategy. The authors are asked to explain this contrasting result.

Regarding the data analysis approach, the employed methodology and the statistical tests adopted are appropriate for the reported aims. The entire procedure is extensively described and enough detailed for allowing its replication.

The authors made all the data underlying the described findings fully available.

Reviewer #2: In this paper, the authors administered a self-reports questionnaire to investigate the prevalence of deception, the characteristics of good-liars and the strategies they used. Results confirmed the previous literature, showing that participants who tell more lies in daily life are those who consider themselves good liars. These individuals mostly lie face to face to colleagues and friends, using verbal strategies like simple, clear and plausible stories that are as much as possible closed to the truth.

Overall, I think that the paper is suitable for this journal. It deals with an interesting topic for the scientific community that study the cognitive and behavioural aspects of deception. The manuscript is well written and clear. However, it could benefit from clarifying some points:

1) To encourage open science and data reproducibility, the authors should publish their data in a repository, so the link to the data should be made available and open.

2) References: I noticed that the number of self-citations is very high. 11 out of 53 references (20%) contain at least one of the authors of the present paper. I invite the authors to reduce self-references.

3) Data are collected through Amazon Mechanical Turk. To addressed the limitations commonly associated with this type of study (these limitations are mentioned by the authors in the manuscript as well, see also http://www.annualreviews.org/doi/abs/10.1146/annurev-clinpsy-021815-093623 and http://journals.plos.org/plosone/article?id=10.1371/journal.pone.0057410#s15), I suggest to expand the sample with additional subjects collected in laboratory. This addition sample can be used to confirm the results obtained from the Turkers.

4) Participants: information about schooling is missed. I think that this is a variable that should not be underestimated. I would like to see a discussion about how this variable may influence the self-reported perception about the ability to lie and the strategy used. If the authors collected this data, I’d like to see some statistics comparing high and low schooling participants and males vs females. For example, are self-reported good liars mostly males with high educational level?

5) Which are the practical implications of this study? How this study enriches the state of the art? And how this study can concretely influence the development of more efficient lie detection machines? Please, discuss this point.

6. PLOS authors have the option to publish the peer review history of their article (what does this mean?). If published, this will include your full peer review and any attached files.

Reviewer #1: No

---

## [Author Response · Author response to Decision Letter 0]

31 Oct 2019

We appreciate the constructive feedback regarding our manuscript entitled “Lie prevalence, lie characteristics and strategies of self-reported good liars”, and the opportunity to address these comments. Please refer to our Response Letter for a detailed overview of our revisions.

---

## [Decision Letter · Decision Letter 1]

8 Nov 2019

Lie prevalence, lie characteristics and strategies of self-reported good liars

PONE-D-19-22731R1

Dear Dr. Verigin,

We are pleased to inform you that your manuscript has been judged scientifically suitable for publication and will be formally accepted for publication once it complies with all outstanding technical requirements.

With kind regards,

Giuseppe Sartori

Academic Editor

PLOS ONE

Additional Editor Comments (optional):

Reviewers' comments:

Reviewer's Responses to Questions

**Comments to the Author**

1. If the authors have adequately addressed your comments raised in a previous round of review and you feel that this manuscript is now acceptable for publication, you may indicate that here to bypass the “Comments to the Author” section, enter your conflict of interest statement in the “Confidential to Editor” section, and submit your "Accept" recommendation.

Reviewer #1: All comments have been addressed

Reviewer #2: All comments have been addressed

2. Is the manuscript technically sound, and do the data support the conclusions?

Reviewer #1: Yes

Reviewer #2: Yes

3. Has the statistical analysis been performed appropriately and rigorously? 

Reviewer #1: Yes

Reviewer #2: Yes

4. Have the authors made all data underlying the findings in their manuscript fully available?

Reviewer #1: Yes

Reviewer #2: Yes

5. Is the manuscript presented in an intelligible fashion and written in standard English?

Reviewer #1: Yes

Reviewer #2: Yes

6. Review Comments to the Author

Reviewer #1: The authors addressed all the comments raised in the previous review phase. In my opinion, although further research in more controlled settings is necessary, the authors' responses were well-argued and provided a reasonable explanation to the raised questions.

Reviewer #2: (No Response)

7. PLOS authors have the option to publish the peer review history of their article (what does this mean?). If published, this will include your full peer review and any attached files.

Reviewer #1: No

Reviewer #2: Yes: Merylin Monaro

---

## [Editor Report · Acceptance letter]

21 Nov 2019

PONE-D-19-22731R1 

Lie prevalence, lie characteristics and strategies of self-reported good liars 

Dear Dr. Verigin:

I am pleased to inform you that your manuscript has been deemed suitable for publication in PLOS ONE. Congratulations! Your manuscript is now with our production department. 

With kind regards,

on behalf of

Dr. Giuseppe Sartori 

Academic Editor

PLOS ONE